# Evaluation of Passive Immunity Induced by Immunisation Using Two Inactivated gE-deleted Marker Vaccines against Infectious Bovine Rhinotracheitis (IBR) in Calves

**DOI:** 10.3390/vaccines8010014

**Published:** 2020-01-04

**Authors:** Stefano Petrini, Cecilia Righi, Carmen Iscaro, Giulio Viola, Paola Gobbi, Eleonora Scoccia, Elisabetta Rossi, Claudia Pellegrini, Gian Mario De Mia

**Affiliations:** 1National Reference Laboratory for Infectious Bovine Rhinotracheitis (IBR), Istituto Zooprofilattico Sperimentale Umbria-Marche “Togo Rosati”, 06126 Perugia, Italy; c.righi@izsum.it (C.R.); c.iscaro@izsum.it (C.I.); p.gobbi@izsum.it (P.G.); e.scoccia@izsum.it (E.S.); e.rossi@izsum.it (E.R.); c.pellegrini@izsum.it (C.P.); gm.demia@izsum.it (G.M.D.M.); 2Veterinary Practitioner, 62026 San Ginesio, Italy; violagiulio@gmail.com

**Keywords:** BoHV-1, marker vaccines, passive immunity

## Abstract

Different types of vaccines against Infectious Bovine Rhinotracheitis (IBR) are commercially available. Among these, inactivated glycoprotein E (gE)-deleted marker vaccines are commonly used, but their ability to induce passive immunity is poorly known. Here, we evaluated the passive immunity transferred from dams immunised with commercial inactivated gE-deleted marker vaccines to calves. We vaccinated 12 pregnant cattle devoid of neutralising antibodies against Bovine alphaherpesvirus 1 (BoHV-1) and divided them into two groups with 6 animals each. Both groups were injected with a different inactivated gE-deleted marker vaccine administrated via intranasal or intramuscular routes. An additional 6 pregnant cattle served as the unvaccinated control group. After calving, the number of animals in each group was increased by the newborn calves. In the dams, the humoral immune response was evaluated before calving and, subsequently, at different times until post-calving day 180 (PCD180). In addition, the antibodies in colostrum, milk, and in serum samples from newborn calves were evaluated at different times until PCD180. The results indicated that inactivated glycoprotein E (gE)-deleted marker vaccines are safe and produce a good humoral immune response in pregnant cattle until calving and PCD180. Moreover, results showed that, in calf serum, passive immunity persists until PCD180.

## 1. Introduction

Bovine alphaherpesvirus 1 (BoHV-1) causes various clinical syndromes among cattle populations, including respiratory disease, vaginitis, balanoposthitis, abortions, enteritis, and encephalitis, which may be observed after acute infection or subsequent to viral recrudescence, following periods of stress [1,2]. To date, Finland, Sweden, Norway, Denmark, Austria, Switzerland, as well as the province of Bolzano in Italy, have successfully employed a “test-and-slaughter” strategy. Other European Union states have implemented compulsory eradication programmes combining “test-and-removal” with “vaccination” using marker vaccines, conversely to vaccination strategies used in the USA, where non-marker vaccines are used [3]. Marker vaccines are derived from the deletion of one or more genes responsible for the synthesis of glycoproteins or enzymes [4,5].

In Europe, glycoprotein E (gE) of BoHV-1 is the gene marker that is most commonly deleted in modified live or inactivated BoHV-1 DIVA (differentiating infected from vaccinated animals) vaccines [6,7]. This type of immunisation makes it possible to differentiate animals immunised with marker vaccines (gE negative) from those infected or inoculated with traditional non-deleted vaccines (gE positive) through diagnostic tests specific for BoHV-1 gE [4,8,9]. However, a disadvantage of using live marker vaccines is that the virus can remain in a latent state in the immunised animals such that it can be reactivated and excreted following exposure to an immunosuppressive stimulus [10]. Further, marker vaccines administered both intramuscularly and intranasally induce a marked humoral and cell-mediated immune responses [11], however, little information is available regarding the induction of passive immunity.

Regarding traditional vaccines, several reports have shown that cattle vaccinated with non-deleted modified-live vaccines transfer neutralising antibodies (NA) to the newborn calves that can protect them from experimental infection [12,13], and other studies have demonstrated that a poor titre of colostral antibodies increases the risk of contracting BoHV-1 infection in the calf [14]. It has also been reported that the concentration of maternal antibodies against BoHV-1 in calves after birth and after colostrum ingestion is directly proportional to that present in the mother’s serum [15,16]. Though extensive studies have reported on marker vaccines, there is a need to assess their ability to induce passive immunity. Here, we have made a concerted effort to evaluate passive immunity transmitted from cattle to newborn calves following immunisation with different inactivated marker vaccines.

## 2. Materials and Methods

### 2.1. Vaccines

Two commercial vaccines were selected for the study (Table 1). The products were identified as follows: vaccine A, a gE-negative, inactivated strain of BoHV-1 containing light paraffin oil; and vaccine B, a gE-negative, inactivated strain of BoHV-1 containing aluminium hydroxide, saponin, and thiomersal. The vaccines were administered as follows: 2 doses of 2 mL each were given to each animal (pregnant females) 20 days apart. Vaccine A was injected to the animals intranasally (i.n.), whereas vaccine B was injected intramuscularly (i.m.) into the neck muscle. The cattle received their first dose of vaccine injection in the 5th month of pregnancy.

### 2.2. Experimental Design

All animals in the study were from a single dairy herd, located in central Italy and owned and run by a local veterinarian. According to the farm records, no vaccine against BoHV-1 had been used before and no recent history of respiratory disease was registered. Eighteen pregnant Friesian cattle of approximately 4–5 years of age and devoid of NA and ELISA antibodies against glycoprotein E (gE-ELISA) of BoHV-1 were selected. The animals had previously had two pregnancies. The status of the animals was confirmed throughout the experimental period based on the absence of clinical symptoms and serological and virological investigation against IBR. The animals were fed twice per day with unifeed mixtures and water *ad libitum*. Moreover, after calving, milking was performed three times per day. The maintenance and experimental protocols were established according to the European legislation on the protection of animals used for scientific purpose [17]. The experiments were carried out with the approval of the Italian Ministry of Health under the authorisation number 653/2018-PR.

The number of animals in each group was determined through the sampling procedure envisaged for an experimental clinical study that was used to compare the proportions in terms of superiority. A percentage of 0 was considered for the proportion of the appearance of the study event in the control group and a percentage of 90 was used as the proportion of the appearance of the study event in the experimental group, with an alpha error of 1% and a study power of 90%.

The cattle were divided into three groups with six pregnant animals each. The first and second group were injected with vaccine A and B, respectively (Table 1), whereas the third group served as an unvaccinated control group. The animals in each group were housed in separate pens on the farm. After calving, the number of animals in each cattle group increased by six newborn calves, and these were fed with 6 L of colostrum or milk administered twice per day throughout the experimental period. Subsequently, 10 days post-birth, the newborn calves were fed with concentrated feed and hay in addition to milk from approximately 10 days after birth.

Data on the following possible adverse reactions after vaccination were obtained from the local veterinarian: (i) localised swelling at the injection site; (ii) slight transient slight increase in body temperature; (iii) anaphylactic shock; (iv) abortion; (v) stillbirth; or (vi) birth of weak, unthrifty calves. In the case of abortion, the foetus and placenta with serum samples were submitted for laboratory investigations. Clinical conditions were monitored throughout the experiment. On the day of vaccination (time 0), day 30, and day 120 post-vaccination days (PVDs), serum samples were collected from the cattle and tested for the presence of BoHV-1 gE-ELISA antibodies and NAs. In addition, serum samples were taken from all animals on 0, 2, 5, 7, 15, 30, 60, 90, 120, 150, and 180 post-calving days (PCDs) to assess BoHV-1 antibodies by gB-ELISA, gE-ELISA, and virus neutralisation test (VNT). Simultaneously, colostrum or milk samples were collected from lactating cattle to test for gB-ELISA, gE-ELISA, or indirect ELISA of BoHV-1.

### 2.3. Sample Collection

Blood samples of approximately 7 mL were collected from the coccygeal or jugular vein of pregnant cattle and calves, respectively. For each animal, disposable needles and vacutainer tubes were used. Subsequently, the samples were centrifuged at 850× *g* for 30 min at 4 °C to extract the serum. In addition, colostrum and milk samples were uniformly collected from the 4 teats of each animal to obtain 50 mL of each using conical tubes. Later, they were centrifuged at 1800× *g* for 30 min at 4 °C to obtain skimmed samples. All samples were transported with refrigeration to the laboratory within 2 h of collection before testing. Afterwards, all samples were stored at −20 °C for further serological studies.

### 2.4. ELISA Tests

Two commercial ELISA tests (IDEXX IBR gE Ab Test, Maine, USA; IDEXX IBR gB X3 Ab, Maine, USA) were used in parallel to examine the collected sera, colostrum, or milk samples. In addition, indirect ELISA (IDEXX BHV1 Bulk Milk Ab, Maine, USA) was used only for colostrum and milk samples. The protocols described by the kit manufacturer were followed and the results were also expressed according to the instructions of the manufacturer. The microplates were read using an automated plate reader and the data were analysed using the Magellan software (Tecan AG, Switzerland).

### 2.5. Neutralisation Test

The serum samples were tested using the protocol described by the OIE Manual of Diagnostic Tests and Vaccines for Terrestrial Animals [18]. Briefly, 50 µL of undiluted serum samples and two-fold dilutions of each were mixed with 50 µL of 100 TCID_50_ of BoHV-1 (Los Angeles strain 01/17) in 96-well microtitre plates. The samples were incubated at 37 °C for 24 h and then 30,000 Madin–Darby Bovine Kidney cells in 100 µL were added to each well. The cells were provided by Biobanking of Veterinary Resources (BVR, Brescia, Italy) and identified with the code BS CL 63. After 4 days of incubation at 37 °C, the plates were read using the inverted tissue culture microscope to determine cytopathic effects. Neutralisation titres were expressed as the highest dilution inhibiting cytopathology.

### 2.6. Statistical Analysis

Overall, 36 animals were used in this study, which included the 18 pregnant cattle and their 18 newborn calves.

The titres of antibodies were measured on a logarithmic scale with base 10. Means of the titres were calculated for each animal group and for all sampling times. The nonparametric Wilcoxon Mann–Whitney test was used to evaluate the presence of any statistically significant differences in immunity induced by vaccination between the two gE-deleted marker vaccines and the unvaccinated controls. The differences between group A and group B with respect to the control group at each sampling time were studied considering a significance level at *P-value* ≤ 0.05. All statistical analyses were performed using Stata software v.11.2 (StataCorp LCC, Texas, USA).

## 3. Results

### 3.1. Clinical Response

After immunisation, no clinical signs or adverse reactions were observed in any of the pregnant cattle or animals immunised i.n. with vaccine A or i.m. with vaccine B. Moreover, throughout the experimental period, no clinical signs of IBR infection were seen in the calves born from cattle immunised with vaccine A or B, except for two calves. These two animals, born to vaccine A-immunised cattle showed mono-lateral discharge at two months of age, and, following a nasal swab for virologic and bacteriologic investigations, were found to be infected with only one *Staphylococcus* spp. Consequently, they were treated with antibiotics (ceftiofur hydrochloride).

### 3.2. Serology

#### 3.2.1. Cattle

After 30 days post-vaccination, all pregnant cattle had NAs to BoHV-1 at a mean NA titre of 2.41 log_10_. Titres in cattle vaccinated with both vaccine A and vaccine B showed a significant difference compared to the control (*P* = 0.0009); the mean titre was 2.41 log_10_. This value was increased to 2.82 log_10_ (*P* = 0.0021; vaccine A) and 3.06 log_10_ (*P* = 0.0013; vaccine B) at PVD120. No seroconversion was detected in the unvaccinated controls (Table 2).

After 2 days post-calving, the mean NA titre of cattle immunised with vaccine A did not decrease compared to that on the day of calving (2.82 log_10_; *P* = 0.0021). However, PCD7 onwards, the NA titres started declining and continued to do so till PCD120 (antibody titres reached 1.76 log_10_; *P* = 0.0021), after which it remained constant until PCD180 (Table 3). Conversely, after 2 days post-calving, the mean NA titres of cattle immunised with vaccine B decreased to 3.01 log_10_ (*P* = 0.0009) and, then, remained constant up to PCD7. Subsequently, they decreased up to PCD180 to an antibody titre of 1.76 log_10_ (*P* = 0.0021). In addition, after vaccination, all pregnant cattle remained seropositive for gB and seronegative for gE. Conversely, no seroconversion was detected in the unvaccinated controls (Table 3).

#### 3.2.2. Newborn Calves

On the day of calving (approximately 7 h after birth), calves born to vaccine A-immunised cattle had NAs with a mean titre of 2.51 log_10_ (*P* = 0.0021), which remained constant up to PCD2. This titre decreased to 2.26 log_10_ (*P* = 0.0020) on PCD7. Subsequently, it declined up to PCD180, reaching 0.85 log_10_ (*P* = 0.0219). Conversely, on the day of calving, the calves of cattle immunised with vaccine B had NAs titre of 2.76 log_10_ (*P* = 0.0017), which persisted till PCD2. It, then, started declining up to PCD180, when antibody titres reached 0.75 log_10_ (*P* = 0.0020). After calving, all calves remained seropositive for BoHV-1 gB and seronegative for BoHV-1 gE. In contrast, no seroconversion was detected in the control group (Table 4).

#### 3.2.3. Colostrum and Milk

The colostrum and milk samples collected from cattle immunised with vaccine A or B were seronegative for gE and seropositive for gB or indirect ELISA antibodies throughout the experimental period. In contrast, the colostrum or milk samples collected from unvaccinated cattle were seronegative for gB, gE, and indirect ELISA antibodies (Table 5).

## 4. Discussion

Currently, in European countries, the DIVA strategy is considered to be one of the first-line interventions in BoHV-1 eradication programs in areas with a high prevalence of the disease [6,9]. In Italy, a national surveillance plan for IBR is only active for autochthonous cattle breeds and recommends the use of marker vaccines to decrease IBR seroprevalence [19]. It is widely known that marker vaccines induce a marked humoral and cell-mediated immune response [11,20]; however, only little information is available regarding passive immunity induced by these vaccines. Previous studies have suggested that passive immunity would be induced in the calves after maternal vaccination. However, this study is important to provide this.

We performed several experiments using two gE-deleted marker vaccines, administered via an intranasal (vaccine A) or intramuscular (vaccine B) route. Both inactivated gE-deleted marker vaccines did not induce any clinical signs or adverse reactions. Such effects, especially abortion and anaphylactic shock, have been reported when modified-live attenuated vaccines are administered to pregnant cattle in the pre-partum period [21,22]. The results of this study, in accordance with those reported previously [22,23], suggested no risk of adverse reactions following the administration of either inactivated gE-deleted marker vaccine. In addition, the outcomes of the present study are in accordance with those of other studies [24], showing that the vaccination of pregnant cattle can prevent IBR-induced abortion. It is currently known [25,26] that other herpesviruses are involved in cattle abortion, but little information is available on the ability of vaccines prepared against BoHV-1 to protect against abortion induced by Bovine gammaherpesvirus 4 (BoHV-4) and Bovine alphaherpesvirus 5 (BoHV-5).

The response to vaccination was determined by the assessment of NA titres against BoHV-1 or by gE-ELISA of serum BoHV-1 in pregnant cattle. Results indicated that NA titres against BoHV-1 were increased at PVD30, compared to control levels. Significantly (*P* = 0.0021; *P* = 0.0013) elevated titres were also observed at PVD120. These findings are in agreement with those of other studies in which an increased colostral antibody production against BoHV-1 and various pathogens were detected after the vaccination of pregnant cattle during the pre-partum period [13,21,22,27]. In contrast, in a study conducted by Lemaire et al., a low level of antibodies against BoHV-1 was observed in two cows after two vaccinations with inactivated gE-deleted marker vaccines [28]. Moreover, in calves, a decline in maternal antibodies against BoHV-1 and Bovine Viral Diarrhea Virus was shown at 168 or 224 days of age, respectively [12,16]. Similar results were also seen in another study, in which a vaccine containing rotavirus, coronavirus, and *Escherichia coli* was used to immunise pregnant cattle and led to high titres of specific antibodies in their colostrum and milk [29]. Further, Agianniotaki et al. demonstrated the presence of NAs against lumpy skin disease virus until 90 days post-calving [30].

In the present study, the pregnant cattle showed negative results for gE-ELISA during the vaccination period. These results are in agreement with those of reports demonstrating that serum samples collected from animals immunised with gE-deleted marker vaccines were also negative for gE-ELISA [4,6,8]. Moreover, this result showed that, during the experimental period, the field virus did not circulate in the experimental groups. Cattle naturally or experimentally exposed to BoHV-1 can be positive for gE-ELISA antibodies [28].

The experiments performed herein showed that pregnant cattle immunised with the two gE-deleted marker vaccines could transfer passive immunity to calves. In fact, on the day of calving, both groups injected with vaccine A or B showed high NA serum levels against BoHV-1. After birth, calves are born in an agammaglobulinemia state and need colostrum with a high level of immunoglobulins (Igs). In the first 24 h, these Igs can penetrate the bloodstream by binding to the Fc receptors located in the intestinal brush border [31,32]. Produced by B cells or plasma cells in the blood of lactating cows [33], these Igs are responsible for the early protection of newborns against different infections. Furthermore, colostrum contains leukocytes and several antimicrobial proteins, such as complement C3, lactoferrin, lactoperoxidase, and lysozyme. These leukocytes, together with maternal Igs, are involved in conferring orogastric protection [34]; specifically, they can enter the circulation through intestinal adsorption to promote neonatal cellular immunity [35].

In our research, on the day of calving, the cattle immunised with vaccines showed a good level of NAs, which decreased progressively up to PCD180. In addition, calves in both the groups showed NA titres similar to those of their mothers approximatively 7 h after birth. Subsequently, the antibody titres decreased until the end of the experiment, when they reached negative values. In both of the vaccinated groups, the difference in antibody titres between cattle and newborn calves could be due to the passage of Igs from mothers to newborns through the colostrum. These results are similar to those obtained by Cervenak et al., showing that when the time of calving approaches, IgG_1_ massively decreases in maternal serum samples as it binds to mammary epithelial cells. Then, IgG_1_ is released with colostrum and milk during lactation. Calves undergo so-called gut closure by approximately 24–48 h after birth and the adsorbed colostral IgG (IgG_1_, IgG_2_) levels fall below 1% [35]. After PCD2, the antibody titres in milk gradually decreased up to PCD180.

The data obtained in this study supported the results of Mechor et al., who reported a correlation in serum NA titres between cattle and newborn calves [13]. In addition, the antibody titres observed in both the groups of newborn calves, born to the cattle immunised with vaccine A or B (0.85 log_10_ and 0.75 log_10_, respectively), on PCD180 were lower than those required to protect them against infection by BoHV-1. Several studies have shown that NAs higher than a value of 0.90 log_10_ can protect calves against experimental infection [5,10,36]. Moreover, it has been shown that these antibodies appear in nasal secretions from calves as early as the first day after the ingestion of colostrum [37]. The NAs secreted in the respiratory tract mucosa, primarily of the IgG_1_ class, persist for 15 to 20 days after birth and serum antibodies can be detected until calves reach several months of age [37]. These results are similar to those of other studies [12] showing the persistence of NAs up to PCD170 when using non-marker vaccines.

This study was carried out based on current field conditions with different variables, including the geographical position of the farm, weather, nutrition, and health status of the herd. Generally, dairy cattle are subjected to stress that can negatively affect their serological responses to vaccination. In this study, there was no evidence of stressors as both intramuscularly- and intranasally-vaccinated animals produced humoral immunity with high antibody titres. Conversely, other authors, using modified-live vaccines, found that the aforementioned factors can negatively affect the antibody response after vaccination [38].

In this study, no challenge with virulent BoHV-1 virus was given because this research aimed to evaluate the ability of the two gE-deleted marker vaccines to induce passive immunity. Moreover, this study was intended to form the basis of additional research evaluating the effect of different vaccines against IBR. Thus, future studies will be conducted to assess whether passive immunity in calves can protect them against experimental infection using virulent BoHV-1 virus. Overall, the two inactivated gE-deleted marker vaccines against BoHV-1 were shown to (i) be innocuous for pregnant cattle, (ii) effectively transfer passive immunity from dams to calves up to PCD180, and (iii) be suitable for immunisation in IBR eradication programs. Further studies will be conducted to evaluate if passive immunity induced by these vaccines can protect calves from challenge with a virulent (*wt*) strain of BoHV-1.

## Figures and Tables

**Table 1 vaccines-08-00014-t001:** Bovine alphaherpesvirus 1 (BoHV-1) gE-deleted marker vaccines used in the experiment.

Vaccine Identification	Type	Adjuvant	Virus Concentration in TCID_50_ in One Dose (2 mL)	Inoculation Route
A	gE-negative, inactivated	Light Paraffin Oil	10 ^5.70^	Intranasal
B	gE-negative, inactivated	Aluminium Hydroxide, Saponin, and Thiomersal	10 ^8.00^	Intramuscular

TCID_50_, Tissue Culture Infectious Dose.

**Table 2 vaccines-08-00014-t002:** Antibody response in serum samples collected from pregnant cattle vaccinated against BoHV-1 using different gE-deleted marker vaccines.

Pregnant Cattle		Post-Vaccination Day (PVD)							
		0		30			120^b^		
No.	Vaccine	gE- ELISA	NA^a^	gE- ELISA	NA	*P-*value^c^	gE- ELISA	NA	*P-*value^c^
6	A	Neg.	<0.30	Neg.	2.41	0.0009	Neg.	2.82	0.0021
6	B	Neg.	<0.30	Neg.	2.41	0.0009	Neg.	3.06	0.0013
6	*Unvaccinated Controls*	Neg.	<0.30	Neg.	<0.30	--	Neg.	<0.30	--

^a^ Antibody titre: mean value; ^b^ The day of calving; ^c^
*P*-value indicating the difference in NA titre between cattle immunised with each vaccine and unvaccinated controls; NA, neutralising antibody.

**Table 3 vaccines-08-00014-t003:** Antibody response in pregnant cattle vaccinated against BoHV-1 using different gE-deleted marker vaccines.

Vaccine	Test	Post-Calving Day (PCD)									
		0	2	7	15	30	60	90	120	150	180
A	gE-ELISA	-	-	-	-	-	-	-	-	-	-
	gB-ELISA	+	+	+	+	+	+	+	+	+	+
	NA ^a^	2.82	2.82	2.61	2.51	2.41	2.16	2.01	1.76	1.76	1.76
	*P*-value ^b^	0.0021	0.0021	0.0021	0.0021	0.0021	0.0020	0.0021	0.0021	0.0021	0.0021
B	gE-ELISA	-	-	-	-	-	-	-	-	-	-
	gB-ELISA	+	+	+	+	+	+	+	+	+	+
	NA	3.06	3.01	3.01	2.86	2.86	2.86	2.76	2.56	2.21	1.76
	*P*-value	0.0013	0.0009	0.0009	0.0017	0.0017	0.0017	0.0019	0.0019	0.0020	0.0021
*Unvaccinated Controls*	gE-ELISA	-	-	-	-	-	-	-	-	-	-
	gB-ELISA	-	-	-	-	-	-	-	-	-	-
	NA	<0.30	<0.30	<0.30	<0.30	<0.30	<0.30	<0.30	<0.30	<0.30	<0.30

^a^ Antibody titre: mean value; ^b^
*P*-value indicating the difference in NA titre between cattle immunised with each vaccine and unvaccinated controls; NA, neutralising antibody.

**Table 4 vaccines-08-00014-t004:** Antibody response in newborn calves derived from pregnant cattle vaccinated against BoHV-1 with different gE-deleted marker vaccines.

Vaccine	Test	Post-Calving Day (PCD)										
		0		2	7	15	30	60	90	120	150	180
A	gE-ELISA		-	-	-	-	-	-	-	-	-	-
	gB-ELISA		+	+	+	+	+	+	+	+	+	+
	NA ^a^		2.51	2.51	2.26	2.11	1.81	1.61	1.20	1.00	0.90	0.85
	*P*-value ^b^		0.0021	0.0021	0.0020	0.0020	0.0021	0.0021	0.0019	0.0073	0.0209	0.0219
B	gE-ELISA		-	-	-	-	-	-	-	-	-	-
	gB-ELISA		+	+	+	+	+	+	+	+	+	+
	NA		2.76	2.76	2.71	2.56	2.46	2.06	1.76	1.20	0.75	0.75
	*P-*value		0.0017	0.0017	0.0017	0.0020	0.0020	0.0019	0.0019	0.0020	0.0020	0.0020
*Unvaccinated Controls*	gE-ELISA		-	-	-	-	-	-	-	-	-	-
	gB-ELISA		-	-	-	-	-	-	-	-	-	-
	NA		<0.30	<0.30	<0.30	<0.30	<0.30	<0.30	<0.30	<0.30	<0.30	<0.30

^a^ Antibody titre: mean value; ^b^
*P*-value indicating the difference in NA titre between cattle immunised with each vaccine and unvaccinated controls; NA, neutralising antibody.

**Table 5 vaccines-08-00014-t005:** Antibody response in colostrum or milk samples collected from cattle immunised against BoHV-1 using different gE-deleted marker vaccines after calving, as measured by ELISA tests.

Vaccine ^a,b^	Test	Post-Calving Day (PCD)										
		0	2	7	15	30	60	90	120	150	180	
A	gE-ELISA^c^	-	-	-	-	-	-	-	-	-	-	
	gB-ELISA^d^	+	+	+	+	+	+	+	+	+	+	
	Indirect ELISA ^e^	+	+	+	+	+	+	+	+	+	+	
B	gE-ELISA	-	-	-	-	-	-	-	-	-	-	
	gB-ELISA	+	+	+	+	+	+	+	+	+	+	
	Indirect ELISA	+	+	+	+	+	+	+	+	+	+	
*Unvaccinated Controls*	gE-ELISA	-	-	-	-	-	-	-	-	-	-	
	gB-ELISA	-	-	-	-	-	-	-	-	-	-	
	Indirect ELISA	-	-	-	-	-	-	-	-	-	-	

^a^ Vaccinated pregnant cattle were housed in separate pens on the farm; ^b^ See Table 1; ^c^ IDEXX IBR gE Ab Test, Maine, USA; ^d^ IDEXX IBR gB X3 Ab, Maine, USA; ^e^ IDEXX BHV1 Bulk Milk Ab, Maine, USA.

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
