# Peer review of "Evaluation of Passive Immunity Induced by Immunisation Using Two Inactivated gE-deleted Marker Vaccines against Infectious Bovine Rhinotracheitis (IBR) in Calves"

_vaccines, 2020, doi:10.3390/vaccines8010014_

Round 1

Reviewer 1 Report

The manuscript describes a trial to study the transfer of antibodies (passive immunity) after vaccination of pregnant dams against IBR with inactivated gE-deleted marker vaccines to their offspring.

The major strength of the manuscript is the very detailed description of the study design, the methods and the results. The tables give all the necessary information.

There are some minor spelling, grammar and phrasing checks required. I have not listed all language issues in the details.

Minor changes:

Abstract:

L23-24: Check phrasing of the sentence “After calving, the pregnant cattle …”.

L25-26: Check content of the sentence: “In addition, passive immunity …”. In my view “passive immunity” refers to a concept of transferring antibodies from one organism to another. Thus, it cannot be measured in colostrum nor milk or other media.

L28: Change “calve” to “calf”.

Introduction:

L33-35: Please rephrase the first sentence to “… among cattle and is responsible for …”.

L35-36: The sentence “To date, European …” is ambiguous, as IBR is nor widespread in Europe and even countries that are free of the virus still vaccinate cattle. Please rewrite.

M&M

L70-71: Please add the information if the cows were pregnant for the first time or had previous pregnancies.

L80-81: Check content of the sentence:” After calving, each group was increased …”.

L88-89: Please write in conditional, as from the figures no abortions did happen.

L89-90: As the vaccination was administered 20 days apart, please specify if the samples were taken at the time of

L100: Change “nipples” to “teats”.

Results

L145-ff and all tables: Concerning the p-values it is not clear to me, which groups you compared. As there is only a single p-value for each group A and B and no for the control group, I assume that you compare the groups with the controls. In the description you describe also a statistical significance test between the groups A and B. Please state explicit which groups you compare in the statistical analysis. However, the statistical test you use is appropriate for samples coming from a not normal distribution but following the same distribution in the two groups that you compare. Therefore, I doubt that the test is appropriate for the comparison of vaccinated groups with the control group, as obviously the distributions of titer values differ between the two groups. I suggest to omit any statistical test between the vaccinated groups and the control group, as the statistical significance does not add value to the study results.

Discussion:

General: Please add to the discussion that the evidence prior to your study did suggest that passive immunity would be induced in the calves after maternal vaccination. However, this study is important to prove this.

L186-193: In this paragraph you refer several times to other studies but without referencing them. Please add the missing references there.

L194-199: Usually bovines are vaccinated when they are not pregnant. Please include in the discussion, if, from your results, vaccination prior to pregnancy would make a difference compared to the protocol applied in this study.

L209-213: Although an interesting information, I feel that this paragraph is too far away from the study objective. Please skip it.

L222-225: Please check the sentence “Moreover, in the vaccinated cattle, the placenta did …”. To me the phrasing is ambiguous, as it suggests that the placenta barrier is a finding of the present study. But it is more or less general knowledge. Additionally, as you did not take a blood sample from the newborn prior to colostrum intake, you did not proof this fact. Please rewrite.

L239-244: I wonder why the titer values of AB did not decrease at the sampling 7days pp? please explain.

Author Response

Review 1

The manuscript describes a trial to study the transfer of antibodies (passive immunity) after vaccination of pregnant dams against IBR with inactivated gE-deleted marker vaccines to their offspring.

The major strength of the manuscript is the very detailed description of the study design, the methods and the results. The tables give all the necessary information.

There are some minor spelling, grammar and phrasing checks required. I have not listed all language issues in the details.

Minor changes:

Abstract:

L23-24: Check phrasing of the sentence “After calving, the pregnant cattle …”.

 Dear reviewer, thank you for your suggestion. The phrasing of the sentence was modified as following: “After calving, the number of animals in each group was increased by the newborn calves.”;

L25-26: Check content of the sentence: “In addition, passive immunity …”. In my view “passive immunity” refers to a concept of transferring antibodies from one organism to another. Thus, it cannot be measured in colostrum nor milk or other media.

Dear reviewer, thank you for your suggestion. The sentence was modified as following: “In addition, the antibodies in colostrum, milk, and in serum samples from the newborn calves were evaluated at different times until PCD180”;

L28: Change “calve” to “calf”.

Dear reviewer, thank you for your suggestion. The term “calve” was changed with “calf”;

Introduction:

 L33-35: Please rephrase the first sentence to “… among cattle and is responsible for …”.

Dear reviewer, thank you for your suggestion. The first sentence was rephrased as following: “Bovine alphaherpesvirus 1 (BoHV-1) causes various clinical syndromes among cattle populations worldwide, including respiratory disease, vaginitis, balanoposthitis, abortions, enteritis, encephalitis, which may be observed after acute infection or viral recrudescence, following periods of stress [1,2]”;

L35-36: The sentence “To date, European …” is ambiguous, as IBR is nor widespread in Europe and even countries that are free of the virus still vaccinate cattle. Please rewrite.

Dear reviewer, thank you for your suggestion. The sentence was rewritten as following: “To date, Finland, Sweden, Norway, Denmark, Austria and Switzerland, as well as the province of Bolzano in Italy, have successfully employed a “test-and-slaughter” strategy. Other EU states have implemented compulsory eradication programmes combining “test and removal” with “vaccination” using marker vaccines, conversely to vaccination strategies used in the USA, where non-marker vaccines are used [3]. Marker vaccines are derived from the deletion of one or more genes responsible for the synthesis of glycoproteins or enzymes [4,5].”;

M&M

 L70-71: Please add the information if the cows were pregnant for the first time or had previous pregnancies.

Dear reviewer, thank you for your suggestion. The following information was added: “The animals had two previous pregnancies”;

L80-81: Check content of the sentence:” After calving, each group was increased …”.

Dear reviewer, thank you for your suggestion. The following information was added: “After calving, the number of animals in each cattle group was increased by six newborn calves”;

L88-89: Please write in conditional, as from the figures no abortions did happen.

 Dear reviewer, thank you for your suggestion. The sentence was modified as following: “Data on following possible adverse reactions after vaccination were obtained from the local veterinarian”;

L89-90: As the vaccination was administered 20 days apart, please specify if the samples were taken at the time of

Dear reviewer, thank you for your suggestion. The sentence was specified as following: “On the day of vaccination (time 0) and day 30 and day 120 post-vaccination (PVDs), serum samples were collected from the cattle and tested for the presence of BoHV-1 gE-ELISA antibodies and NAs”;

L100: Change “nipples” to “teats”.

Dear reviewer, thank you for your suggestion. The term “nipples” was changed to “teats”;

Results

L145-ff and all tables: Concerning the p-values it is not clear to me, which groups you compared. As there is only a single p-value for each group A and B and no for the control group, I assume that you compare the groups with the controls. In the description you describe also a statistical significance test between the groups A and B. Please state explicit which groups you compare in the statistical analysis.

 However, the statistical test you use is appropriate for samples coming from a not normal distribution but following the same distribution in the two groups that you compare. Therefore, I doubt that the test is appropriate for the comparison of vaccinated groups with the control group, as obviously the distributions of titer values differ between the two groups. I suggest to omit any statistical test between the vaccinated groups and the control group, as the statistical significance does not add value to the study results.

Dear reviewer, thank you for your suggestion. In the statistical analysis section we compared each group vs control group. The following sentence was added in Tables 2,3 and 4: “P-value indicating the difference in NA titre between cattle immunised with each vaccine and the unvaccinated controls”. About the use of Wilcoxon–Mann–Whitney test, as you rightly say, it is appropriate for samples coming from a not normal distribution but following the same distribution in the two groups that you compare, both symmetrical or asymmetrical. So, we choose the W to determine and to confirm the significance of the results and because the assumption of the test is that samples follow the same distributions symmetry or approximate symmetry is less restrictive than assumption of normality or approximate normality. What state above is reported in the following books: 1) Individual comparisons by ranking method, In Biometrics, 1, pp. 80-83; 2) On a test of whether one of two random variables is stochastically larger than the other, Annals of Mathematical Statistics, 18, pp. 50-60; 3) Historical notes on the Wilcoxon unpaired two-sample test, Journal of the American Statistical Association, 52, pp. 356-360.

 Discussion:

 General: Please add to the discussion that the evidence prior to your study did suggest that passive immunity would be induced in the calves after maternal vaccination. However, this study is important to prove this.

Dear reviewer, thank you for your suggestion. In the discussion section we inserted the following phrase: “Previous studies have suggested that passive immunity would be induced in the calves after maternal vaccination. However, this study is important to prove this”;

 L186-193: In this paragraph you refer several times to other studies but without referencing them. Please add the missing references there.

 Dear reviewer, thank you for your suggestion. In this paragraph, references were added;

 L194-199: Usually bovines are vaccinated when they are not pregnant. Please include in the discussion, if, from your results, vaccination prior to pregnancy would make a difference compared to the protocol applied in this study.

 Dear reviewer, thank you for your insightful suggestion. We are unable to incorporate the suggested text in our manuscript as we do not have any experimental evidence for it. However, the vaccines used in this study are registered by the European Medicines Agency for use even during pregnancy. Thus, we believe that the results obtained in this study should not be different from those involving vaccination before pregnancy; however, this needs validation.

 L209-213: Although an interesting information, I feel that this paragraph is too far away from the study objective. Please skip it.

 Dear reviewer, thank you for your suggestion. In the discussion section we eliminated the following paragraph: “In veterinary medicine, the practice of maternal vaccination to obtain newborn protection against various infectious diseases has been in use for a long time. In human medicine, in the last few years, this practice has been re-evaluated after several outbreaks of pandemic influenza and pertussis. In particular, in pregnant women, the USA and EU health authorities recommended the use of inactivated vaccines rather than live vaccines against herpesviruses (varicella and zoster) [31].”

 L222-225: Please check the sentence “Moreover, in the vaccinated cattle, the placenta did …”. To me the phrasing is ambiguous, as it suggests that the placenta barrier is a finding of the present study. But it is more or less general knowledge. Additionally, as you did not take a blood sample from the newborn prior to colostrum intake, you did not proof this fact. Please rewrite.

 Dear reviewer, thank you for your suggestion. In the discussion section we rewritten the following paragraph: “After birth, calves are born in the agammaglobulinemia state and need colostrum with a high level of immunoglobulins (Igs). In the first 24 h, these Igs can penetrate the bloodstream by binding to the Fc receptors located in the intestinal brush border [31-32].

 L239-244: I wonder why the titer values of AB did not decrease at the sampling 7days pp? please explain.

Dear reviewer, thank you for your suggestion. We think that AB titers did not decrease at 7 days pp because the concentration of antibodies transmitted by dams to calves was high throughout the experiment period.

Reviewer 2 Report

AUTHORS

Manuscript ID: vaccines-625240

Title: Evaluation of passive immunity induced by immunisation with two inactivated gE-deleted marker vaccines against infectious bovine rhinotracheitis (IBR) in calves

This is a study evaluating passive immunity transferred to calves via maternal immunisation with commercial inactivated gE-deleted marker vaccines. Pregnant cattle were vaccinated to Bovine alphaherpesvirus 1 (BoHV-1, inactivated gE-deleted marker vaccine) via intranasal or intramuscular routes. The humoral immune response was evaluated in the dams. Passive immunity was evaluated in colostrum, milk, and in serum samples of the newborn calves. This is a truly interesting study and deserves to be published, after minor clarifications are made.

Authors state that eighteen pregnant Friesian cattle of approximately 4–5 years of age were devoid of NA and ELISA antibodies to glycoprotein E. Please describe how was this status confirmed.

I am far from being an expert on ethics protocols, particularly those regulated by the Italian legislation, but authors state that experiments were carried out under the approval of the Italian Ministry of Health. Shouldn’t this approval be obtained from the Agricultural governmental sections?

When first describing the vaccine administration, please clarify further details “injected with vaccine A, the second group was inoculated with vaccine B” where was the injection, dose, inoculation, etc.

Section 2.6. Statistical analysis could benefit from exclusion of sample size calculation (number of animals to be used) and placing of this information on 2.2. Experimental design

On 3.1. “… bacteriologic investigations, only one Staphylococcus spp. was detected….” Place spp. without italic

Authors claim that there was no evidence of “stressors as both intramuscularly- and intranasally-vaccinated animals produced humoral immunity with high antibody titres.”. I don’t believe this is strong evidence of absence of stress. No cortisol or other stress indicators where measured to assume

Author Response

Review 2

This is a study evaluating passive immunity transferred to calves via maternal immunisation with commercial inactivated gE-deleted marker vaccines. Pregnant cattle were vaccinated to Bovine alphaherpesvirus 1 (BoHV-1, inactivated gE-deleted marker vaccine) via intranasal or intramuscular routes. The humoral immune response was evaluated in the dams. Passive immunity was evaluated in colostrum, milk, and in serum samples of the newborn calves. This is a truly interesting study and deserves to be published, after minor clarifications are made.

Authors state that eighteen pregnant Friesian cattle of approximately 4–5 years of age were devoided of NA and ELISA antibodies to glycoprotein E. Please describe how this status was confirmed.

- Dear reviewer, thank you for your suggestion. The sentence was modified as following “The status of the animals was confirmed throughout the experimental period based on negative clinical symptoms and serological and virological investigation against IBR”.

I am far from being an expert on ethics protocols, particularly those regulated by the Italian legislation, but authors state that experiments were carried out under the approval of the Italian Ministry of Health. Shouldn’t this approval be obtained from the Agricultural governmental sections?

Dear reviewer, thank you for your suggestion. In Italy, the scientific research of Veterinary Medicine is under the approval of the Ministry of Health.

When first describing the vaccine administration, please clarify further details “injected with vaccine A, the second group was inoculated with vaccine B” where was the injection, dose, inoculation, etc.

Dear reviewer, thank you for your suggestion. This information is added in the following paragraph: “The vaccines were administered as follows: two doses of 2 ml each were given to each animal (pregnant females) 20 days apart. Vaccine A was injected to the animals intranasally (i.n.), whereas vaccine B was injected intramuscularly (i.m.) into neck muscle.”  

Section 2.6. Statistical analysis could benefit from exclusion of sample size calculation (number of animals to be used) and placing of this information on 2.2. Experimental design

Dear reviewer, thank you for your suggestion. We believe that the sample size calculated statistically is important for the study. For this reason, we moved this information on 2.2. Experimental design section;

On 3.1. “… bacteriologic investigations, only one Staphylococcus spp. was detected….” Place spp. without italics

Dear reviewer, thank you for your suggestion. The change was made;

Authors claim that there was no evidence of “stressors as both intramuscularly- and intranasally-vaccinated animals produced humoral immunity with high antibody titres.”. I don’t believe this is strong evidence of absence of stress. No cortisol or other stress indicators where measured to assume

Dear reviewer, thank you for your suggestion. Cortisol or other stress indicators were not investigated in this study because they were not in the experimental design. However, it is known that in dairy cattle stress causes a reduction in milk production. In our study, the cattle never showed a decrease in their milk production. Associating high production of milk (around 30 liters / day) with high production of immunity, it can be assumed that the animals were not subjected to stress.